# The Longitudinal Relationship between Edentulism and the Progress of Multimorbidity

**DOI:** 10.3390/nu16142234

**Published:** 2024-07-11

**Authors:** Rolla Mira, Jonathon Timothy Newton, Wael Sabbah

**Affiliations:** Faculty of Dentistry, Oral & Craniofacial Sciences, King’s College London, London SE1 9RT, UK; rolla.mira@kcl.ac.uk (R.M.); tim.newton@kcl.ac.uk (J.T.N.)

**Keywords:** ageing, edentulism, multimorbidity, nutrient intake, longitudinal studies

## Abstract

Objectives: To examine the longitudinal relationship between edentulism, nutritional intake, and the progress of multimorbidity among older Americans. Methods: We used data from the Health and Retirement Study (2006–2018), a longitudinal survey of older Americans that has collected data biennially since 1992. Edentulism was assessed in 2006 while nutritional intake was assessed in 2013. Multimorbidity was indicated by five self-reported chronic conditions: diabetes, heart conditions, lung diseases, cancer, and stroke. Individuals with two or more conditions at baseline were excluded from the analysis. Nutritional intake was calculated by summing 10 nutrients (protein, vitamins C, D, B12 and E, calcium, zinc, polyunsaturated fatty acids, folate, and ß-carotene). Structural equation modelling (SEM) was used to examine the nutritional pathway between edentulism (2006) and the increase in multimorbidity from 2006 to 2018. Results: The number of individuals included in the analysis was 3463. The incidence of multimorbidity between 2006 and 2018 was 24.07%, while the percentage of edentate participants in 2006 was 16.42%. The mean total nutrition in 2013 was 4.50 (4.43, 4.55). The SEM analysis showed that edentulism was negatively associated with nutritional intake {estimate −0.15 (95%CI: −0.30, −0.01)}. A negative association was found between total nutrition and multimorbidity {estimate −0.008 (95%CI: −0.01, −0.002)}. Age, wealth, and smoking were included in the analysis and had statistically significant associations with multimorbidity. Conclusion: The analysis demonstrated a longitudinal association between edentulism, nutritional intake, and the progress of multimorbidity.

## 1. Introduction

The elderly population of the world is rapidly expanding [1]. According to the WHO, one in six individuals on the planet will be 60 years of age or older by 2030 [2]. At this point, there will be 1.4 billion people over the age of 60, up from 1 billion in 2020. The number of individuals in the world who are 60 or older is expected to double (to 2.1 billion) by 2050. It is anticipated that between 2020 and 2050, the number of people 80 years of age or older will triple, reaching 426 million [2]. Furthermore, the development of multimorbidity is significantly influenced by ageing, with a frequency in older adults ranging from 55 to 98% [3]. Multimorbidity is known as the existence of two or more chronic diseases in one person. Because of its negative effects on health and society, multimorbidity is becoming a major healthcare concern [4,5]. Approximately 45% of Americans, or 133 million people, have at least one chronic illness, and the percentage is rising [6]. Hospitalisation, long-term impairment, lower quality of life, and even mortality can result from chronic diseases, which include cancer, diabetes, hypertension, stroke, heart disease, respiratory disorders, obesity, arthritis, and oral diseases [7].

Tooth loss is a significant global public health issue [8]. Based on the Centers for Disease Control and Prevention (CDC), one in six persons aged 65 years and older in the USA are edentulous [9]. Moreover, in the United States, half of all individuals between the ages of 20 and 64 have experienced at least one case of permanent tooth loss [10]. Furthermore, the National Health and Nutrition Examination Survey (NHANES) data from 2009 to 2014 revealed that 6.2 million Americans aged 50 and older, 17.6% of adults aged 65 and older, and 22.5% of those aged 75 and older in the US were affected by edentulism [11]. 

The dental and medical fields have mostly acknowledged the oral–systemic relationship in recent years. New research has been conducted and continues to demonstrate the relationship between dental and systemic health. For instance, the primary cause of tooth loss, periodontal disease, was linked via the inflammatory process to several systemic disorders [12,13,14]. 

Furthermore, in terms of individual chronic conditions, tooth loss was associated with certain chronic conditions such as diabetes [15] and cardiovascular diseases [16], while in terms of multimorbidity, Felton [17] found a significant association between tooth loss and multimorbidity. 

Besides multimorbidity, tooth loss negatively affects people’s dietary intake. Being edentate was significantly associated with a lower consumption of dietary fibre, vitamin C, and other nutrients [18,19]. Moreover, two studies found associations between tooth loss, obesity, and malnutrition [20,21]. On the other hand, inadequate nutrition is an important determinant of multimorbidity [22,23].

People who have lost all of their teeth may experience low self-esteem and social isolation as a result of their inability to communicate and engage with others in daily life [24]. Common risk factors, such as dietary deficiencies, alcoholism, and smoking, which are all connected to low socioeconomic status, may have an impact on the association between tooth loss and multimorbidity [25,26].

The current study aims to examine the relationship between edentulism, nutritional intake, and the progress of multimorbidity among older Americans.

## 2. Study Design

Longitudinal data from the Health and Retirement Study (HRS) were used in this study. With response rates consistently above 80%, data were collected every two years since 1992. Participants received study materials prior to each interview. Oral consent was obtained right before each interview, and participants were given a confidentiality statement. Since 2006, half of the panel respondents have additionally participated in an upgraded face-to-face interview. Data collection was conducted by a researcher employed by the HRS. The questionnaires were standardised to reduce risk of bias. The current work did not require ethical approval because the HRS data are publicly accessible and do not contain personal information.

### 2.1. Study Sample

Adult Americans 50 years of age and older made up the study sample [27]. Since the introduction of the oral health question started in 2006, data from that year were included in the current analysis. Inclusion criteria included: those over 50 years who responded to the oral health question in 2006, had no more than one chronic condition in 2006, and took part in the 2013 nutritional module. Individuals who lacked complete data for any of the characteristics covered in each wave were not included in the analysis.

### 2.2. Study Variables

The predictor variables for this study were divided into three main categories: demographic factors, health-related behaviours, and socioeconomic factors. The demographic variables were gender (male, female) and age, whereas socioeconomic position (SEP) was indicated by total wealth and its measures were reported in nominal dollars and calculated as the sum of the appropriate wealth components minus debt. In this analysis we used wealth in 2006 and categorised it into quartiles (highest, second highest, second lowest, and lowest).

Smoking in 2006 was selected as a behavioural factor, and it was categorised as a dichotomous variable (never/former smoker and current smoker).

Complete tooth loss (edentulism) in 2006 was indicated by answering “yes” to the question “Have you lost all your upper and lower natural permanent teeth?”.

The Health and Retirement Study included one file where individual nutritional intake was calculated in 2013. This wave is called the Health Care and Nutrition Study (HCNS) 2013. Individual intake was calculated according to the values of nutrient recommendations: dietary reference intake (DRI). According to the available nutrients in the HRS, only 10 out of 13 nutrients of the US Food and Drug Administration were chosen: protein, vitamin C, vitamin D, vitamin B12, vitamin E, calcium, zinc, polyunsaturated fatty acids, folate, and ß-carotene. These nutrients were calculated based on the DRI [28] as the following: protein (men < 56 g/day; women < 46 g/day); vitamin C (men 90 mg/day; women 75 mg/day); vitamin D (both 50–70 years, 15 µg/day; both >70 years, 20 µg/day—the HRS calculated vitamin D in international units, which contradicted the US Food and Drug Administration, who calculated it in micrograms, so there are two measures for vitamin D according to age. Based on this, a convertor was used to accurately measure the values, yielding 15 IU = 600 and 20 IU = 800); vitamin B12 (<2.4 µg/day); vitamin E (15 mg/day); zinc (men < 11 mg/day; women < 8 mg/day); polyunsaturated fatty acids (men 160 mg; women 90 mg); folate (<400 µg/day); ß-carotene (3 mg).

Each of these nutrients was coded as 0 = inadequate intake or 1 = adequate intake and were summed up in one variable, total nutrients, ranging from 0 to 10.

Multimorbidity was indicated by self-reported diagnoses of five chronic conditions: diabetes, heart conditions, lung diseases, cancer, and stroke. These conditions were selected because they are highly prevalent among older adults and are classified as major causes of disability and death in the USA [9]. In this analysis, multimorbidity in 2018 was used as a dichotomous variable: 1 or fewer condition versus 2 or more conditions. Participants with more than 1 chronic condition at baseline (2006) were excluded from the analysis.

The final analysis included data on multimorbidity, tooth loss, nutritional intake, wealth, smoking, age, and gender. These data were from Health and Retirement Survey, specifically from RAND HRS Longitudinal File 2018 (V1).

This study was human observational and followed properly the STROBE guidelines.

## 3. Statistical Analysis

The incidence of multimorbidity among older Americans aged 50 years and over in 2018 was included. The prevalence of edentulism in 2006 and the mean total nutrition (2013) were also included. The analysis was limited to people with complete data. The variable with most missing cases was nutrition since it was obtained from a one-off survey of a subsample of the HRS.

First, we assessed the distribution of multimorbidity (2018), tooth loss (2006), and total nutrition (2013). We also assessed tooth loss in 2012 and 2018; however, they were not included in the analysis as tooth loss in 2012 is unlikely to have an impact on nutrition in 2013, and the tooth loss in 2018 occurred after nutritional assessment in 2013.

To test the relationship between total tooth loss and multimorbidity through nutritional intake, structure equation modelling (SEM) was used. One path analysis diagram was created with multiple adjustments. The following associations were tested: (1) associations of wealth and smoking with tooth loss adjusting for age and gender; (2) association between tooth loss and total nutrition adjusting for age and gender; (3) associations between total nutrition and multimorbidity adjusting for smoking, wealth, and age. All analyses were conducted using Stata 14.2

## 4. Results

The analysis included 3463 participants. Figure 1 shows the flowchart of the number of eligible participants from 2006 to 2018. The incidence of multimorbidity in 2018 was 24.07%. The percentage of participants with complete tooth loss in 2006 was 10.11%. The mean total nutrition in 2013 was 4.50 (95%CI: 4.43, 4.55) (Table 1). In 2012 and 2018, the percentages of edentates were 12.5% and 16.7%, respectively. Figure 2 demonstrates the direction of association between tooth loss in 2006, total nutrition in 2013, and multimorbidity in 2018. Those who lost all their teeth in 2006 had a significantly lower score of adequate nutrition in 2013 {estimate −0.15 (95%CI: −0.30, −0.01)}. A negative association was observed between adequate nutrition in 2013 and multimorbidity {estimate −0.008 (95%CI: −0.01, −0.002)} after accounting for smoking and wealth (Table 2).

## 5. Discussion

In this study, we examined the relationship between edentulism and progress of multimorbidity among older adults in the United States using longitudinal data from the HRS and examined the role of nutritional intake in this relation. The analysis using SEM demonstrated that nutritional intake is in the pathway between edentulism and multimorbidity. Tooth loss in 2006 was negatively associated with adequate total nutritional in 2013, which in turn was negatively associated with the incidence of multimorbidity in 2018. These results were consistent with previous studies reporting that tooth loss was associated with declines in fruit, vegetable, and fibre consumption, which in turn influenced the risk of chronic conditions [29,30].

Tooth loss affects chewing ability and limits the selection of food items, which impacts the adequate intake of essential nutrients [18,19]. For example, older adults with fewer teeth or no teeth will have difficulties eating fresh fruits and vegetables and could consume other unhealthy food [18,31]. While this relationship was always clear when tooth loss was indicated by a number of teeth or functional dentition, it was less clear when complete edentulism was used; however, this was mainly attributed to the lack of longitudinal studies [21]. The current analysis found a longitudinal association between complete tooth loss and inadequate nutritional intake. On the other hand, there is evidence of a consistent relationship between inadequate nutrition and a higher incidence of multimorbidity and comorbidity [22,23]. The impact of nutrition on the onset and management of chronic illnesses has been widely acknowledged [23], which has led to recommendations for food consumption based on daily reference intakes for particular macro- and micronutrients [22]. The findings of the current analysis highlight the role of adequate nutritional intake in the progress of multimorbidity even after accounting for other behavioural and socioeconomic factors.

In this study, we did not test the direct relationship between tooth loss in 2006 and incidence multimorbidity in 2018 as the focus of the research question was on demonstrating the indirect relation through inadequate nutritional intake. However, there are several studies that demonstrated association between tooth loss and multimorbidity [32], and between tooth loss and individual chronic conditions [16,20,33,34]. It is worth noting that in the studied sample, complete tooth loss slightly increased in 2012 and 2018, but there was no point in including tooth loss in these two waves in the analysis as there would be a very short time span to show the impact of tooth loss in 2012 on nutritional intake in 2013. Furthermore, tooth loss in 2018 occurred after the nutritional assessment.

While the nutritional pathway between tooth loss and multimorbidity was tested in this study, there are other pathways for this relationship, including a psychological pathway due to the impact of tooth loss on talking and socialising with others, which could lead to social isolation [35], impacting multimorbidity. Adverse socioeconomic factors and associated risk behaviours could also have a common impact on tooth loss and multimorbidity [11,36].

The strength of this study is in using longitudinal data of older American adults to assess the relationship between tooth loss, nutritional intake, and multimorbidity, and in using structural equation modelling to demonstrate the nutritional pathway. The nutritional intake variable was based on the recommendations for essential nutrient intake, which affects chronic conditions. The focus on the progress of multimorbidity and tooth loss, both common public health problems in the ageing population, is another strength of this analysis. There are a few limitations worth mentioning. First, we used self-reported doctor diagnoses of chronic conditions, which could be subject to recall bias. However, including clinical assessments is very difficult and expensive in longitudinal studies of this size. Oral health was limited to self-reported complete tooth loss, which is also subject to recall bias. Furthermore, there were data on the use of dentures, or the condition of dentures, which could affect the ability to eat in this analysis. A better indicator of oral health would have been to include the number of teeth or functional dentition and the use of dentures. However, we made the best use of the available longitudinal data, which included repeated measures of comprehensive medical data to demonstrate the longitudinal association between tooth loss, nutritional intake and the incidence of multimorbidity. Finally, there were some attritions of the sample, but this is common in surveys of older adults, which included data collected over 12 years. Furthermore, most of the exclusion was due to a lack of data on nutrition, which was collected from a subsample of the population.

The findings of this study on the impact of tooth loss on nutritional intake and the progress of multimorbidity highlight the importance of identifying health promotion interventions to improve oral health and nutritional intake to halt the progress of multimorbidity among older adults. Future research should include an intervention study design to examine the relationship demonstrated in this study.

## 6. Conclusions

This study demonstrated a longitudinal association between tooth loss and inadequate nutritional intake and between nutritional intake and the progress of multimorbidity, using a national longitudinal survey of older American adults. The study highlights the importance of both oral health and nutritional intake in the progress of multimorbidity among older American adults.

## Figures and Tables

**Figure 1 nutrients-16-02234-f001:**
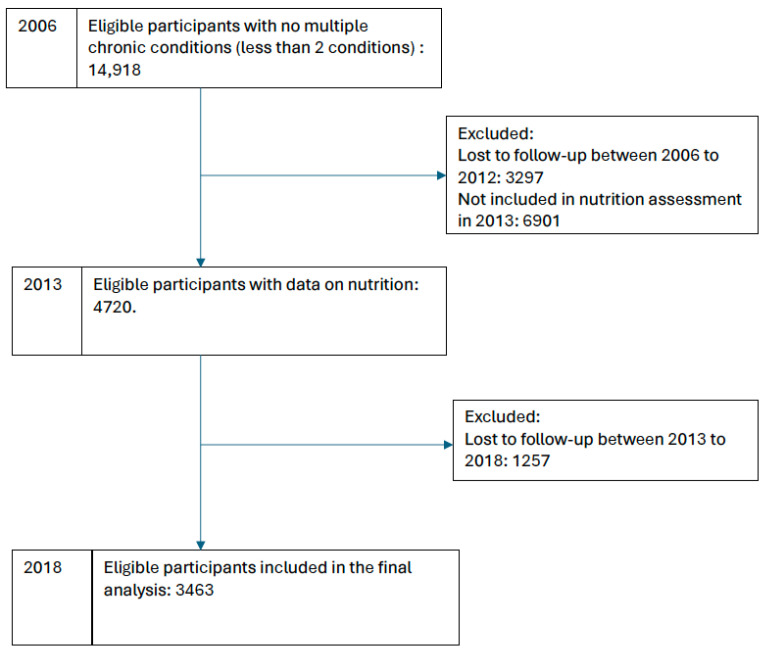
Flowchart of the number of participants included in the analysis from 2006 to 2018.

**Figure 2 nutrients-16-02234-f002:**
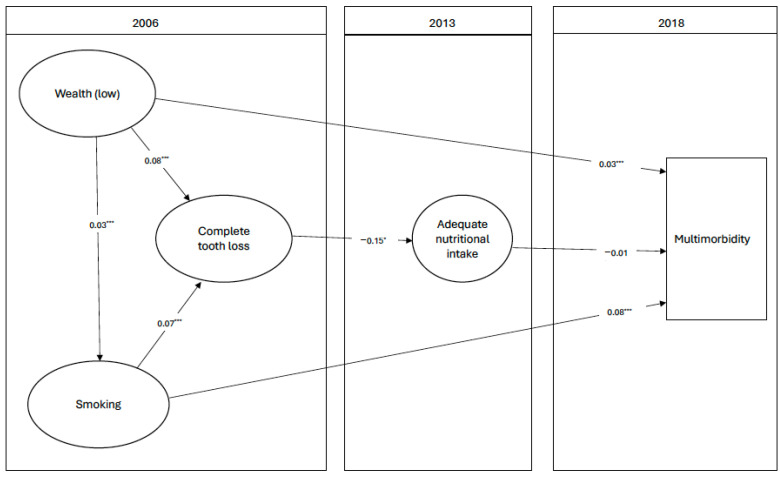
Nutritional pathway between edentulism (2006) and multimorbidity (2018). * *p* < 0.05, *** *p* < 0.001. All associations were adjusted for age. Participants with 2 or more chronic conditions in 2006 were excluded.

**Table 1 nutrients-16-02234-t001:** Characteristics of participants, 2006–2018; distribution of variables (exposures and outcomes) (N: 3463).

Variables		Mean/Percentages
Age (mean) ^1^		63.36 (63.08, 63.65)
Gender ^1^	Male	37.79%
	Female	62.21%
Tooth loss in 2006 ^1^	Dentate	88.98%
	Edentate	10.11%
Current/former smokers 2006		51.8%
Wealth quartiles	Highest	34.9%
	2nd highest	29.1%
	2nd lowest	23.6%
	Lowest	12.4%
Total nutrition in 2013 (mean) ^1^		4.50 (4.43, 4.55)
Multimorbidity in 2018 ^2^	1 condition	75.93%
	2 conditions or more	24.07%

^1^, Exposure variables; ^2^, Outcome.

**Table 2 nutrients-16-02234-t002:** Path coefficients estimated using a structural equation model for the relationship between tooth loss, nutritional intake, and multimorbidity.

Standardised		Coefficient	Confidence
Smoking	Age	−0.01 ***	−0.01, −0.02
	Lower wealth 2006	0.03 ***	0.02, 0.03
Edentulism 2006	Age	0.007 ***	0.006, 0.008
	Gender	0.01	−0.001, 0.02
	Lower wealth 2006	0.08 ***	0.07, 0.08
	Smoking 2006	0.07 ***	0.06, 0.08
Total nutrition 2013	Age	0.01	−0.01, 0.01
	Gender	0.37 ***	0.26, 0.48
	Edentulism 2006	−0.15 *	−0.30, −0.01
Multimorbidity 2018	Age	0.008 ***	0.007, 0.01
	Total nutrition 2013	−0.008 *	−0.015, −0.002
	Lower wealth 2006	−0.03 ***	−0.01, −0.04
	Smoking 2006	0.08 ***	0.05, 0.11

* *p* < 0.05, *** *p* < 0.001.

## Data Availability

Data used in this study is available in: https://hrs.isr.umich.edu/about accessed on 5 June 2024.

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
