# Peer review of "The Longitudinal Relationship between Edentulism and the Progress of Multimorbidity"

_nutrients, 2024, doi:10.3390/nu16142234_

Round 1
Reviewer 1 Report
Comments and Suggestions for Authors
The article entitled "The Relationship between Edentulism and the Progress of Multimorbidity" presents significant and contemporary clinical relevance. The authors analyzed data from a sample of American patients, focusing on oral health, systemic comorbidities, and information about 10 nutrients. The findings are interesting, demonstrating a longitudinal association between edentulism, nutritional intake, and the progression of multimorbidity. However, several methodological and design issues must be addressed to clarify potential misunderstandings and improve the manuscript.
While the authors claim adherence to the STROBE guidelines, the checklist was not provided. Several items do not comply with these guidelines. A major concern is the study design, as the authors fail to specify the type of study in the title or abstract. Additionally, there is no flowchart detailing patient follow-up from eligibility to the end of the study. The authors do not mention the number of patients lost during follow-up, which could introduce significant bias affecting the study's representation and external validity. Furthermore, the inclusion and exclusion criteria are not clearly stated according to STROBE guidelines. There is also no information on which researchers were involved in data collection or how bias control was managed concerning blinding. This information is crucial to prevent omissions and enhance bias control.
The timing of data collection (2006, 2013, and 2018) is justified in the discussion as necessary to analyze the indirect relationship through inadequate nutritional intake, but it remains an irreversible limitation. Despite understanding the study's objective, it should not prevent the authors from presenting baseline and follow-up data for all variables at each time point. This is particularly important as a patient initially self-declared as dentate may experience further tooth loss over the years. Moreover, self-declaration of edentulous does not provide adequate information about the type of oral rehabilitation used or its establishment, nor does it affirm the patient's oral functionality. Consequently, it is not precise to correlate this variable with others in terms of the functionality of the stomatognathic system and nutrients. Including more detailed information on oral conditions, if feasible, would be highly recommended.
Additionally, some results were scarcely discussed, such as the sociodemographic characteristics of the study sample in Table 1 and the habits presented in Table 2. This could undermine the external validity of the study population compared to other studies.
The discussion of the study's limitations appears superficial, given the methodological issues of this magnitude. Addressing these concerns would significantly enhance the study's credibility and impact.
Comments on the Quality of English LanguageThe English is of good quality, but some terms could be revised.
Author Response
Thank you for your constructive critique of the paper.
The article entitled "The Relationship between Edentulism and the Progress of Multimorbidity" presents significant and contemporary clinical relevance. The authors analyzed data from a sample of American patients, focusing on oral health, systemic comorbidities, and information about 10 nutrients. The findings are interesting, demonstrating a longitudinal association between edentulism, nutritional intake, and the progression of multimorbidity. However, several methodological and design issues must be addressed to clarify potential misunderstandings and improve the manuscript.
While the authors claim adherence to the STROBE guidelines, the checklist was not provided. Several items do not comply with these guidelines.
Response: we have now provided STROBE checklist as an additional file
A major concern is the study design, as the authors fail to specify the type of study in the title or abstract.
Response: we have now specified the study design in the title and abstract (longitudinal study).
Additionally, there is no flowchart detailing patient follow-up from eligibility to the end of the study. The authors do not mention the number of patients lost during follow-up, which could introduce significant bias affecting the study's representation and external validity.
Response: We have now included a flowchart of the eligible participants from 2006 to 2018 (Figure 1). The majority of those excluded from the study were due to lack of data on nutrition which was collected from a subsample of the survey. There were also some attritions which is expected in this type of longitudinal data of older adults over 12 years. We have commented on this in the discussion.
Furthermore, the inclusion and exclusion criteria are not clearly stated according to STROBE guidelines.
Response: we have elaborated on the inclusion and exclusion criteria in the method section
There is also no information on which researchers were involved in data collection or how bias control was managed concerning blinding. This information is crucial to prevent omissions and enhance bias control.
Response: We have included additional information on data collection in the methods. Data collection was carried out by researchers working with the Health and Retirement Study. The questionnaires were standardised by HRS. Given that the survey collects data on various health and social issues, was no specific research question on mind, those collecting the data were not blinded.
The timing of data collection (2006, 2013, and 2018) is justified in the discussion as necessary to analyze the indirect relationship through inadequate nutritional intake, but it remains an irreversible limitation. Despite understanding the study's objective, it should not prevent the authors from presenting baseline and follow-up data for all variables at each time point. This is particularly important as a patient initially self-declared as dentate may experience further tooth loss over the years.
Response: We have included information on edentulism in 2012 and 2018 as the reviewer requested. However, additional cases of edentulism in 2012 are unlikely to have an impact on nutritional intake in 2013. Additional cases of edentulism in 2018 occurred after assessment of nutrition in 2018. Furthermore, Structural Equation Modelling would not be ideal to assess repeated measures of exposure. We have elaborated on these points and their limitation in the discussion.
Moreover, self-declaration of edentulous does not provide adequate information about the type of oral rehabilitation used or its establishment, nor does it affirm the patient's oral functionality. Consequently, it is not precise to correlate this variable with others in terms of the functionality of the stomatognathic system and nutrients. Including more detailed information on oral conditions, if feasible, would be highly recommended.
Response: we agree with the reviewer on this point, but we could not include additional information on use of denture or replacement of teeth due to data limitation. However, availability of detailed oral health assessments in longitudinal health surveys is very rare, and we made the best of the available data. We have included a discussion of this in the limitations of the study.
Additionally, some results were scarcely discussed, such as the sociodemographic characteristics of the study sample in Table 1 and the habits presented in Table 2. This could undermine the external validity of the study population compared to other studies.
Response: we have no included data on behavioural and socioeconomic factors in Table 1
The discussion of the study's limitations appears superficial, given the methodological issues of this magnitude. Addressing these concerns would significantly enhance the study's credibility and impact.
Response: we have elaborated in the discussion of the limitations.
Reviewer 2 Report
Comments and Suggestions for Authors
This study demonstrated a longitudinal association between tooth loss and inadequate nutritional intake and between nutritional intake and progress of multimorbidity, using national longitudinal survey of American older adults. The study highlights the importance of both oral health and nutritional intake in the progress of multimorbidity among older American adults. A well written paper that highlights the need for dental and medical workers to co-operate more in the future!
One comment and one question: Page 1, line 42. The abbreviation CDC should be explained! The RDA reference (28) seems to be from 2005. No update later??https://ods.od.nih.gov/HealthInformation/Dietary_Reference_Intakes.aspx
Author Response
Response to reviewer 2
Thank you for your constructive critique of the paper.
This study demonstrated a longitudinal association between tooth loss and inadequate nutritional intake and between nutritional intake and progress of multimorbidity, using national longitudinal survey of American older adults. The study highlights the importance of both oral health and nutritional intake in the progress of multimorbidity among older American adults. A well written paper that highlights the need for dental and medical workers to co-operate more in the future!
One comment and one question: Page 1, line 42. The abbreviation CDC should be explained!
Response: We have now explained the abbreviation CDC
The RDA reference (28) seems to be from 2005. No update later?? https://ods.od.nih.gov/HealthInformation/Dietary_Reference_Intakes.aspx
Response: Thank for this comment. In fact, this is the latest available data on recommendation for dietary intake
Round 2
Reviewer 1 Report
Comments and Suggestions for Authors
The authors responded adequately to the suggestions and clarified the most concerning points. However, the article has limitations related to design and methodological conditions, such as those concerning the use of prostheses and oral rehabilitation, which were noted in the discussion. Additionally, they did not present all data on the outcomes assessed throughout the entire longitudinal period of the study. In my opinion, this omission can be reconsidered consideration regardless of the timing of data loss and the collection of nutritional analysis data. Despite these limitations, I believe the article can contribute valuable information to the current literature.